# Multi-Omics Analyses Reveal Mitochondrial Dysfunction Contributing to Temozolomide Resistance in Glioblastoma Cells

**DOI:** 10.3390/biom13091408

**Published:** 2023-09-19

**Authors:** Huaijin Zhang, Yuling Chen, Xiaohui Liu, Haiteng Deng

**Affiliations:** MOE Key Laboratory of Bioinformatics, Center for Synthetic and Systematic Biology, School of Life Sciences, Tsinghua University, Beijing 100084, China; zhang-hj16@mails.tsinghua.edu.cn (H.Z.); chenyuling2016@mail.tsinghua.edu.cn (Y.C.); xiaohui2013@mail.tsinghua.edu.cn (X.L.)

**Keywords:** glioblastoma, temozolomide, chemoresistance, mitochondrial dysfunction, mitochondrial retrograde signaling

## Abstract

Glioblastoma (GBM) is the most common and aggressive malignant brain tumor with poor prognosis. Temozolomide (TMZ) is the standard chemotherapy for glioblastoma treatment, but TMZ resistance significantly compromises its efficacy. In the present study, we generated a TMZ-resistant cell line and identified that mitochondrial dysfunction was a novel factor contributing to TMZ resistance though multi-omics analyses and energy metabolism analysis. Furthermore, we found that rotenone treatment induced TMZ resistance to a certain level in glioblastoma cells. Notably, we further demonstrated that elevated Ca^2+^ levels and JNK–STAT3 pathway activation contributed to TMZ resistance and that inhibiting JNK or STAT3 increases susceptibility to TMZ. Taken together, our results indicate that co-administering TMZ with a JNK or STAT3 inhibitor holds promise as a potentially effective treatment for glioblastoma.

## 1. Introduction

Glioblastoma (GBM) is the most common glioma in adults, accounting for 51% of primary malignant brain tumors [1]. It is classified as a World Health Organization (WHO) grade IV tumor based on histopathological features and affects both children and adults with a slight predominance in males [2]. Glioblastoma is one of the most aggressive tumors, with patients having a median survival of 10–12 months and a 5-year survival rate below 2% following diagnosis [3].

Current treatment options for glioblastoma include maximal safe tumor resection, concurrent chemotherapy, and radiation [4]. The first-line chemotherapy agent used for GBM is temozolomide (TMZ). It has been used for decades, and no drugs have yet exceeded the therapeutic effect of it [5]. TMZ is a monofunctional DNA alkylating agent that induces apoptosis to kill tumor cells. However, clinical observations have shown that the sensitivity of GBM to TMZ treatment decreases over time. This may be closely related to multiple inherent or acquired mechanisms that confer resistance to chemotherapy [6]. Various internal factors such as DNA repair [7,8], epigenetic modifications [9,10], and ABC transports [11] were proven to induce chemoresistance. Molecular mechanisms underlying TMZ resistance have been previously investigated, involving DNA repair systems like O-6-methylguanine-DNA methyltransferase (MGMT) [8,12], metabolic reprogramming [13], and glioblastoma stem-like cells (GSCs) [14].

Mitochondria are cellular organelles associated with the regulation of cellular metabolism, redox signaling, energy generation, regulation of cell proliferation, and apoptosis [15]. Alterations in mitochondrial function impact cellular metabolism and critically influence whole-body health and even life span [16]. A growing number of studies have shown the connection between mitochondrial dysfunction and diseases, such as diabetic kidney disease (DKD) [17], neurodegenerative diseases [18], and cardiovascular disease [19].

Mitochondria generate a wide range of retrograde signals, through which they regulate various cellular activities and protect against mitochondrial dysfunction by activating the expression of nuclear genes. Depending on the stimuli, retrograde signaling can be classified into energetic stress responses, Ca^2+^-dependent responses, and reactive oxygen species (ROS) stress responses [20]. In mammalian cells, impaired mitochondria cause elevated cytosolic free Ca^2+^, which activates Ca^2+^-dependent protein kinases, thereby activating different nuclear transcription factors. And these transcription factors orchestrate multiple biological processes, encompassing cell proliferation and apoptosis [21,22,23].

In this study, we constructed a TMZ-resistant glioblastoma cell line and performed proteomic and metabolomic analyses of the cells. In addition, we performed energy metabolism analysis, which together with proteomic and metabolomic analyses suggested mitochondrial dysfunction alongside activation of the JNK–STAT3 pathway in TMZ-resistant glioblastoma cells. Furthermore, we demonstrated that the inhibition of JNK and STAT3 increased the susceptibility of TMZ-resistant glioblastoma cells to TMZ treatment. These findings provide a novel insight into the mechanism of TMZ resistance and a new therapeutic strategy for glioblastoma patients with TMZ resistance.

## 2. Materials and Methods

### 2.1. Cell Culture

Human glioblastoma cell line U87 WT (U87 wild type) was purchased from the Cell Bank of the Chinese Academy of Sciences (Shanghai, China). The TMZ-resistant cell line U87 DR (U87 drug resistance) was generated from U87 WT treated with increasing concentrations of TMZ. U87 WT and U87 DR cells were cultured in Dulbecco’s Modified Eagle Medium (DMEM) (Wisent, Nanjing, China) supplemented with 1% penicillin/streptomycin (Wisent, Nanjing, China) and 10% fetal bovine serum (FBS) (Wisent, Nanjing, China). Cells were cultured in a 37 °C humidified incubator containing 5% CO_2_.

### 2.2. Cell Counting Kit-8 (CCK-8) Assay

For cell proliferative ability examination, cells were seeded in 96-well plates at 1000 cells/well. Cell proliferation rate was examined with a CCK-8 assay kit (ApexBio, Houston, USA) according to the manufacturer’s instructions. CCK-8 reagent was added into wells at different time points (0, 24, 48, and 72 h after cell inoculation). After 1.5 h of incubation at 37 °C, absorbance at 450 nm was measured using a microplate reader (Thermo Fisher Scientific, Waltham, MA, USA).

For detecting survival rates of U87 WT and U87 DR cells treated with TMZ, cells were seeded into 96-well plates with 5000 cells/well. After 12 h of incubation, TMZ (0, 10, 20, 50, 100, 200, 500, and 1000 μM) was added into complete DMEM medium at different concentrations. After an additional 48 h of incubation, the cell viability was examined using a CCK-8 assay as described above.

### 2.3. Colony Formation Assay

Cells were seeded in 6-well plates at 500 cells/well and incubated for 7 days at 37 °C. After washing twice with PBS, cells were fixed with 1 mL of 100% methanol for 20 min at room temperature and rinsed with water. Then, 500 μL of 0.1% (*w*/*v*) crystal violet staining solution was added into each well for 5 min at room temperature. Colony formation was examined using a bright-field microscope and recorded using Image Lab 6.0.0.25 (Bio-Rad Laboratories, Hercules, CA, USA). Colony coverage was calculated using Fiji Java8 software (Fiji, NIH, Bethesda, MD, USA).

### 2.4. Cell Cycle Analysis

Cells in 6 cm plates were harvested, washed twice with PBS, and fixed with ice-cold 70% ethyl alcohol. After 2 h of fixing at 4 °C, cells were centrifuged, resuspended in 50 μL of RNase A (TIANGEN, Beijing, China), and incubated at 37 °C for 30 min. Then, 400 μL of propidium iodide (LEAGENE, Beijing, China) was added and the mixture was incubated for 30 min before flow cytometry analysis (BD, FACSCalibur, Franklin Lakers, NJ, USA).

### 2.5. Quantitative Real-Time PCR (qPCR)

Total RNA was extracted from cells with Trizol reagent (TIANGEN, Beijing, China). cDNA was synthesized from 2 μg total RNA using a reverse transcription kit (VAZYME, Nanjing, China) according to the manufacturer’s instructions. Total DNA was isolated from cells using a DNA Extraction Kit (TIANGEN, Beijing, China) according to the manufacturer’s instructions. qPCR was performed using the Roche LightCycler 96 System (Roche, Basel, Switzerland) with SYBR green reaction mixture (APExBIO, Houston, TX, USA) following the manufacturer’s instructions. ACTB served as an internal control for mRNA quantification. mtDNA copy number was measured by the relative abundance of NADH dehydrogenase subunit 1(ND1), cytochrome oxidase 1 (COⅠ), and cytochrome oxidase 2 (COⅡ) with ACTB (nuclear genome) serving as an internal control. Primers were designed with Primer-Blast (www.ncbi.nlm.nih.gov, accessed on 13 January 2021), and primer sequences are provided in Appendix A.

### 2.6. Western Blotting Analysis

Cells were resuspended in RIPA (radio-immunoprecipitation assay) lysis buffer (Solarbio, Beijing, China) supplemented with protease inhibitor cocktail (Selleck, Darmstadt, Germany). Then, cells were sonicated for 2 min, followed by centrifugation to collect the supernatant. Protein concentrations were measured using a BCA (bicinchoninic acid) protein assay kit (Solarbio, Beijing, China). Equal amounts of proteins were separated by sodium dodecyl sulfate–polyacrylamide gel electrophoresis (SDS–PAGE) and transferred onto PVDF (polyvinylidene fluoride) membranes (Millipore, Billerica, MA, USA) and subjected to standard Western blot analysis. Anti-JNK, anti-phospho-JNK (Thr183/Tyr185), anti-STAT3, and anti-phospho-STAT3 (Ser727) antibodies were purchased from Cell Signaling Technology (Danvers, MA, USA). Anti-caspase-3 and anti-caspase-3 (activated) antibodies were purchased from Sangon Biotech (Shanghai, China). Anti-β-actin antibody was purchased from ABclonal Technology (Wuhan, China). Anti-mouse and anti-rabbit secondary antibodies were purchased from Cell Signaling Technology (Danvers, MA, USA). Except for the anti-β-actin antibody, which was diluted in 1:10,000, all antibodies were diluted in 1:2000.

### 2.7. Quantitative Proteomics Analysis

Proteins were extracted from cells with 8 M urea in phosphate-buffered saline (PBS) (Wisent, Nanjing, China) containing 1×protease inhibitor cocktail. Then, cells were sonicated for 2 min and centrifugated to collect the supernatant. Protein concentrations were measured using a BCA protein assay kit. A total of 100 μg of protein extracted from each cell line was reduced with 10 mM dithiothreitol (DTT) at room temperature and alkylated with 40 mM iodoacetamide (IAM) in the dark at room temperature. Then, proteins were diluted to 1.5 M urea with PBS and digested with trypsin (Promega, Madison, WI, USA) overnight at 37 °C. The tryptic peptides were desalted using Sep-Pak desalting columns (Waters, Milford, MA, USA) and then desalted peptides were labeled with tandem mass tag (TMT) 6-plex reagents (Thermo Fisher Scientific, Waltham, MA, USA). The combined TMT-labeled peptides were desalted by Sep-Pak columns and separated on a UPLC 3000 system (Thermo Fisher Scientific, Waltham, MA, USA) with an XBridge^TM^ BEH300 C18 column (Waters, Milford, MA, USA) at a flow rate of 1 mL/min. The mobile phase A was H_2_O (pH 10) and the mobile phase B was acetonitrile (pH 10). Peptides were separated with a gradient elution including an increase from 8% to 18% phase B for 30 min and from 18% to 32% phase B for 22 min. Forty-eight fractions were collected and dried by speedvac and recombined into twelve fractions. The fractions were dissolved in 20 μL of 0.1% (*v*/*v*) formic acid (FA) and analyzed by LC–MS/MS.

### 2.8. Liquid Chromatography–Tandem Mass Spectrometry (LC–MS/MS) Analysis

Labeled peptides were separated with a high-performance liquid chromatography (HPLC) system (Thermo Fisher Scientific, Waltham, MA, USA), which was connected to a Q Exactive HFX mass spectrometer (Thermo Fisher Scientific, Waltham, MA, USA). The mass spectrometer used the data-dependent acquisition mode with the Xcalibur 3.0 software (Thermo Fisher Scientific, Waltham, MA, USA). The parameters were as follows: a single full-scan mass spectrum was obtained in the Orbitrap (300–1800 *m*/*z*, 60,000 resolution) and the automatic gain control (AGC) target was 3 × 10^6^; acquisition settings for MS/MS spectra were 45,000 for resolution with an AGC target of 1 × 10^5^ and maximum injection time of 100 ms; the isolation window width was 0.4 Da; and the normalized collision energy for dissociation was 35%.

### 2.9. Peptide and Protein Identification

The UniProt human database (19 October 2019; reviewed; 20,302 sequences) was used by the Sequest HT in Proteome Discoverer (PD) 2.3 software (Thermo Fisher Scientific, Waltham, MA, USA) to search the MS/MS spectra. The searching process used the following setting: full tryptic specificity required, tolerance was two missed cleavages, fixed modifications were carbamidomethylation and TMT 6-plex, variable modification was oxidation, tolerance was 10 ppm for precursor ion mass for all MS acquired in the Orbitrap mass analyzer, and fragment ion mass tolerance was 20 mmu for all MS2 spectra acquired in the Orbitrap. The searched data were further processed with the percolator function in Proteome Discoverer to filter with 1% peptide false discovery rate (FDR). Relative protein quantification was performed using PD 2.3 based on the intensities of reporter ions per peptide.

### 2.10. Metabolomics Analysis

Metabolomics analysis was performed as previously described [24]. Briefly, the cells were washed twice with cold PBS and incubated with pre-chilled 80% methanol (−80 °C) for 1 h at −80 °C. Then, the cells were scraped in 80% methanol on dry ice and centrifuged for 5 min. The extracted metabolites in the supernatant were dried using a lyophilizer and the protein concentrations of pellets were measured for normalization. The dried metabolites were dissolved in 80% methanol and analyzed by LC–MS/MS. We used the TSQ Quantiva Triple Quadrupole Mass Spectrometer (Thermo Fisher Scientific, Waltham, MA, USA) with positive/negative ion switching for the quantitative analysis of targeted metabolites. The Q-Exactive Mass Spectrometer (Thermo Fisher Scientific, Waltham, MA, USA) was chosen for untargeted metabolites profiling. Metabolites were identified based on accurate ion masses and MS/MS fragments. Relative quantitation of metabolites was analyzed with TraceFinder 3.2 (Thermo Fisher Scientific, Waltham, MA, USA).

### 2.11. Oxidative Phosphorylation Analysis

Oxygen consumption rates (OCR) were measured using the Seahorse XF Cell Mito Stress Test using the XF96 Flux Analyzer (Agilent Technologies, Waldbronn, Germany). Cells were seeded at 12,000 cells/well and incubated in a cell incubator with 5% CO_2_ at 37 °C for 16 h. On the day of the assay, cell culture medium was removed and replaced by assay medium supplemented with reagents according to the manufacturer’s instructions. For OCR measurement, 1 μM ATP synthase inhibitor oligomycin, 2 μM uncoupler fluoro-carbonyl cyanide phenylhydrazone (FCCP), and 1 μM complex III inhibitor antimycin A were sequentially injected. The mitochondrial respiration was analyzed and visualized using Wave software (version 2.3.0, Seahorse Bioscience, Agilent Technologies, Waldbronn, Germany) and normalized to the cell number.

### 2.12. Mitochondrial Membrane Potential Measurement

Cells in 6 cm plates were harvested and washed three times with pre-cooled PBS, followed by resuspension in staining buffer containing JC-10 (Solarbio, Beijing, China) according to manufacturer’s instructions. After incubation for 20 min at room temperature, the percentages of MMP reduced were analyzed by using flow cytometry (BD, FACSCalibur, USA) within 1 h. And the absorbance intensities of fluorescence at Ex./Em. = 515/529 nm and Ex./Em. = 585/590 nm were measured.

### 2.13. Cytosolic Calcium Ion Measurement

For content of free cytosolic Ca^2+^ in U87 WT and U87 DR cells, cells in 6 cm plates were harvested and washed three times with pre-cooled PBS. Then, cells were lysed in lysis buffer (cat. S1063S-1, Beyotime, Beijing, China) on ice, followed by centrifugation to collect the supernatant. An amount of 50 μL of supernatant was mixed with 150 μL working buffer. After incubation for 15 min at room temperature in the dark, the absorbance at 575 nm was measured.

To determine the relative content of Ca^2+^ in U87 WT cells with rotenone treatment, cells in 6 cm plates were harvested and washed three times with PBS, followed by resuspension in staining buffer containing 2 μM Fluo-4 AM (Beyotime, Beijing, China). After incubation for 30 min at 37 °C, the relative contents of free cytosolic Ca^2+^ were analyzed by using flow cytometry (BD, FACSCalibur, USA). And the fluorescence at Ex./Em. = 494/516 nm was measured.

### 2.14. Cell Apoptosis Analysis

Cells were seeded in 6 cm plates at 1 × 10^6^ cells/mL. After 24 h of culture, cells were harvested and washed three times with pre-cooled PBS, followed by resuspension in binding buffer containing PE annexin V and 7-AAD (BD Biosciences, San Jose, CA, USA) according to the manufacturer’s instructions. After incubation for 15 min in the dark at room temperature, the percentages of apoptosis were analyzed by using flow cytometry (BD, FACSCalibur, USA) within 1 h. And the fluorescence intensities at Ex./Em. = 496/575 nm and Ex./Em. = 546/650 nm were measured.

### 2.15. Caspase-3 Activity Assay

Cells in 6 cm plates were harvested and washed three times with pre-cooled PBS. Then, cells were lysed in lysis buffer (Cat. C1115-1, Beyotime, Beijing, China) on ice for 15 min, followed by centrifugation to collect the supernatant. Subsequently, 50 μL of supernatant was mixed with 150 μL working buffer and 10 μL Ac-DEVD-pNA (2 mM). Following a two-hour incubation at 37 °C, the absorbance at 405 nm was measured using a microplate reader (Thermo Fisher Scientific, Waltham, MA, USA).

### 2.16. Mitotracker Staining Assay

Cells in 6 cm plates were harvested and washed three times with PBS, followed by resuspension in pre-warmed PBS containing 200 nM Mitotracker dye (Invitrogen, Waltham, MA, USA). After incubation for 30 min at 37 °C, the fluorescence at Ex./Em. = 579/599 nm was detected by flow cytometry. 

### 2.17. Statistical Methods

Statistical data analysis was performed with Origin 2018 software (OriginLab, Northampton, MA, USA). *p* values were generated using unpaired two-tailed Student’s *t*-tests. *p* values lower than 0.05 were considered significant.

The research workflow is summarized in Appendix A.

## 3. Results

### 3.1. Cell Proliferation Was Decreased in TMZ-Resistant Glioblastoma Cells

We generated a TMZ-resistant cell line (U87 DR (U87 drug resistance)) by the stepwise selection of the glioblastoma cell line U87 (U87 WT) cultured in growth media with increasing drug concentrations (Figure 1a). The viability of U87 WT and U87 DR cells treated with different concentrations of TMZ for 48 h was examined by CCK-8 assay. The results showed that U87 DR cells were less sensitive to TMZ treatment than U87 WT cells, with an IC50 of 2273 μM, which is 35 times higher than the IC50 of U87 WT cells, which was 64 μM (Figure 1b and Appendix A).

TMZ is known to induce cell apoptosis in the process of cell proliferation. Thus, CCK-8 and colony formation assays were performed to evaluate the proliferation of TMZ-resistant glioblastoma cells. The number of U87 DR cells was half that of U87 WT cells at 48 h and only a quarter at 72 h (Figure 1c). Similarly, the colony formation assay showed that the cell density of U87 WT cells reached ~44%, while that of U87 DR cells only reached ~12% after 7 days of culture (Figure 1d). The results suggest that the proliferation of U87 DR cells was significantly lower than that of U87 WT cells.

Accordingly, we performed cell cycle analysis by flow cytometry, which revealed G1-phase cell cycle arrest in U87 DR cells, as featured by the increased proportion of G1-phase cells (Figure 1e). Quantitative PCR experimentation was performed to examine the mRNA levels of cyclins and cyclin-dependent kinases (CDKs) involved in G1/S phase transition. The results showed that the mRNA levels of cyclin-dependent kinase 2 (CDK2), cyclin-dependent kinase 4 (CDK4), cyclin-dependent kinase 6 (CDK6), cyclin D1, cyclin E, cyclin A2, and cyclin B were lower in U87 DR cells than in U87 WT cells (Figure 1f).

### 3.2. Apoptotic Process Was Inhibited in TMZ-Resistant Glioblastoma Cells

Apoptosis is induced by TMZ treatment to kill glioblastoma cells, and inhibition of apoptosis is one of the mechanisms of drug resistance. Cell apoptosis analysis was performed using a two-fluorophore flow cytometric method. Phycoerythrin-annexin V (PE annexin V) and 7-aminoactinomycin D (7-AAD) were used to differentiate viable cells (live), early apoptotic cells (apoptotic), and late apoptotic and necrotic cells (dead). After 48 h of treatment with 200 μM TMZ, the percentages of cells in each apoptotic stage were examined. The results showed that 22.8% and 4.2% of U87 WT cells and U87 DR cells, respectively, were in the early apoptotic stage, that is, PE annexin V-positive and 7-AAD-negative. Similarly, 11.3% of U87 WT cells and 1.1% of U87 DR cells were observed to be PE annexin V- and 7-AAD-double positive. These results indicated that a higher portion of U87 WT cells were in early and late apoptotic stages upon TMZ treatment compared with U87 DR cells (Figure 2a). Hence, the cell apoptotic process was inhibited in U87 DR cells to reduce cell death caused by TMZ.

Caspase 3 is a major regulator of the apoptotic process [25]. Next, we measured caspase-3 activity in U87 WT and U87 DR cells under basal conditions and after treatment with 200 μM TMZ. Under basal conditions, a lower caspase-3 activity was observed in U87 DR cells, which was about 30% of the caspase 3 activity in U87 WT cells. When treated with 200 μM TMZ, the caspase-3 activity in U87 DR cells was only 6.7% of that in U87 WT cells (Figure 2b). In line with this, Western blotting results showed that the protein expression level of active caspase 3 (cleaved form) was decreased in U87 DR cells when treated with 200 μM TMZ (Figure 2c). These results suggest that U87 DR had a lower activity of caspase 3 than U87 WT cells, which reduced the apoptotic process and prevented cell death when the glioblastoma cells acquired drug resistance.

### 3.3. Proteins Involved in Glycolysis and TCA Cycle Changes were Differentially Expressed in U87 WT and U87 DR Cells

Next, to explore the mechanism of how glioblastoma cells acquire drug-resistance, we performed quantitative proteomics analysis of U87 WT and U87 DR cells. Quality control analysis of proteomics data was first performed. All data obey a normal distribution and Pearson’s correlation coefficient is greater than 0.89 between experimental groups (Appendix A).

The percentage variations corresponding to 88% coverage were taken as the threshold cut-off [26]; thus, proteins with ratios ≥ 1.1 or ≤0.9 and Student’s *t*-test *p* values < 0.05 were regarded as upregulated or downregulated proteins, respectively (Figure 3a). A total of 6646 proteins were identified in biological triplicates including 1516 upregulated proteins and 1370 downregulated proteins in U87 DR cells compared with U87 WT cells (Figure 3b). Subsequently, gene ontology (GO) analysis of differentially expressed proteins (DEPs) between U87 WT cells and U87 DR cells was performed using the Database for Annotation Visualization and Integrated Discovery (DAVID).

The biological processes of upregulated proteins were significantly enriched in intracellular transport, autophagy, and glycolysis (Figure 3c). In contrast, proteins related to cell cycle, chromatin organization, and TCA cycle were downregulated (Figure 3d). It was particularly noteworthy that the expression of proteins involved in energy metabolism were changed significantly in U87 DR cells. The expression of key kinases in the glycolytic pathway, such as hexokinase (HK), 6-phosphofructokinase (PFK), phosphoglycerate kinase (PGK), and pyruvate kinase (PK), were increased by 1.2-fold or higher than those in U87 WT cells (Appendix A). The expression of lactate dehydrogenase (LDH) was also increased, which implies an altered lactate level in U87 DR cells. Conversely, the expression of proteins involved in the TCA cycle, such as aconitate hydratase (ACO), isocitrate dehydrogenase 2 (IDH2), succinyl-CoA ligase (SUCL), and succinate dehydrogenase (SDH), were downregulated (Appendix A), suggesting the TCA cycle pathway was inhibited in U87 DR cells compared with U87 WT cells.

In addition, autophagy was upregulated in U87 DR cells which has been reported to contribute to drug resistance by inhibiting the apoptotic process [27,28]. To examine if autophagy contributes to TMZ resistance in U87 DR cells, we treated the cells with either an agonist and or an inhibitor of autophagy. CCK-8 assay results showed that the activation or inhibition of autophagy did not significantly change the susceptibility of U87 WT cells or U87 DR cells, respectively, to TMZ-induced cell death (Appendix A). Thus, these results highlight the importance of alterations in glycolysis and TCA cycle pathways in the acquisition of TMZ resistance in U87 cell lines.

### 3.4. Metabolomic Analysis Reveale Changes Involved in Glycolysis and TCA Cycle in U87 DR Cells

In the results of proteomic analysis, we observed the expression of proteins involved in glycolysis were higher, whereas proteins in TCA cycle were lower in U87 DR cells. To verify the changes in these pathways, we next performed metabolomic analysis of U87 WT and U87 DR cells in five biological replicates. The results showed that lactic acid level was increased in U87 DR cells, whereas other intermediates in glycolysis pathway either had little change or were reduced (Figure 4a,b). The elevated protein expression levels and lactic acid level together suggest accelerated glycolysis in U87 DR cells.

Additionally, metabolomic analysis results showed that metabolites involved in the TCA cycle were reduced in U87 DR cells, including cis-aconitic acid, isocitrate, succinic acid, fumaric acid, and malic acid. This is consistent with the observations in the proteomic analysis results showing that enzymes involved in the TCA cycle, such as aconitate hydratase, isocitrate dehydrogenase, succinate-CoA ligase, and succinate dehydrogenase, were downregulated (Appendix A), indicating that the TCA cycle was inhibited in U87 DR cells. Notably, acetyl-CoA, which is the oxidized product of glycolysis and, subsequently, feeds into the TCA cycle, was significantly increased in U87 DR cells (Figure 4c,d). This may be caused by the enhanced glycolysis and inhibited TCA cycle in U87 DR cells. Proteomics and metabolomics results together reveal metabolic reprogramming in U87 DR cells compared with U87 WT cells.

### 3.5. Mitochondrial Function Was Disturbed When Glioblastoma Cells Acquired TMZ Resistance

Energy metabolism in U87 WT and U87 DR cells was further investigated. Mitochondrial oxidative phosphorylation was measured by examining the oxygen consumption rate (OCR) using a Seahorse flux analyzer. The results showed that the basal respiration and spare respiration capacity were markedly decreased in U87 DR cells. Additionally, proton leak, ATP production, and maximal respiration were also decreased (Figure 5a–c and Appendix A). These results suggest a lower basal respiration capacity and ATP production in U87 DR cells. In addition, decreased proton leak suggests a damaged electron transport chain (ETC), causing reduced energy production in U87 DR cells.

Mitochondria are semi-autonomous organelles, and their own genome encodes proteins involved in ETC complexes (except for complex Ⅱ) which are crucial to mitochondrial function [29]. Chronic TMZ treatment could damage mitochondrial DNA (mtDNA) due to the lack of a DNA repair mechanism. Thus, quantitative PCR was performed to assess mtDNA copy number by calculating the ratio of mtDNA to nDNA. Lower relative contents of *MT-CO1*, *MT-CO2*, and *MT-ND1* were detected in U87 DR cells compared with U87 WT cells (Figure 5d), which indicates that mtDNA copy number was reduced in TMZ-resistant glioblastoma cells. Quantitative PCR results also showed that the mRNA levels of transcription factors (NRF1, PGC-1α, and TFAM) that regulate mtDNA replication were reduced in U87 DR cells (Appendix A), whereas Mitotracker staining results showed that the numbers of mitochondria were not significantly different between the two cell lines (Appendix A).

Next, we measured mitochondrial membrane potential (MMP, ∆Ψm), an indicator of mitochondrial activity, using JC-10 staining. The results showed that ∆Ψm was lower in U87 DR cells than in U87 WT cells (Figure 5e), suggesting the depolarization of MMP in U87 DR cells. All these results indicate the existence of mitochondrial dysfunction in TMZ-resistant glioblastoma cells. In line with this, treatment resistance was reportedly observed in mtDNA-depleted glioblastoma cells [30]. Therefore, we next used rotenone, an inhibitor of ETC, to induce mitochondrial dysfunction in U87 WT cells. CCK-8 assay results showed that rotenone treatment reduced the sensitivity of U87 WT cells towards TMZ treatment when the cells were treated with 1 μM rotenone and different concentrations of TMZ simultaneously for 48 h (Figure 5f). However, the survival rate of U87 WT cells treated with rotenone remained lower than that of U87 DR cells when the concentration of TMZ was higher than 200 μM. These results suggest that the acquired TMZ resistance was partly due to mitochondrial dysfunction in glioblastoma cells.

### 3.6. JNK–STAT3 Pathway Was Activated in TMZ-Resistant Glioblastoma Cells

Mitochondrial dysfunction generates various retrograde signals to regulate the expression of nuclear genes [20]. Calcium ions (Ca^2+^) served as retrograde signals, and we observed a higher level of calcium ions in U87 DR cells than in U87 WT cells (Figure 6a). Elevated Ca^2+^ levels directly activate Ca^2+^-regulated kinases, such as MAPK and JNK. Western blotting assays revealed increased phosphorylation levels of JNK, suggesting that JNK was activated in U87 DR cells. However, the phosphorylation levels and expression levels of MAPK showed no significant changes in U87 DR cells (Figure 6b). Next, we treated cells with SP600125, a JNK inhibitor, along with different doses of TMZ. We observed that inhibition of JNK reduced the survival rate of U87 DR cells upon TMZ treatment (Figure 6c).

Furthermore, we observed that phosphorylation levels of Ser727 in STAT3, the downstream target of JNK, were higher in U87 DR cells than in U87 WT cells (Figure 6d). Meanwhile, SP600125 treatment reduced STAT3 Ser727 phosphorylation levels in U87 DR cells (Figure 6e). Next, we examined expression levels of STAT3 downstream anti-apoptotic proteins, including Bcl-xL and survivin, in U87 WT and U87 DR cells. qPCR results showed that mRNA levels of Bcl-xL were increased in U87 DR cells (Figure 6f). Similarly, inhibition of STAT3 using NSC74859 sensitized U87 DR cells to TMZ treatment (Figure 6g). These results suggest that NK–STAT3 pathway activation contributes to acquired TMZ resistance.

## 4. Discussion

Current therapies for GBM patients require the combination of chemotherapy and radiotherapy. Resistance to conventional therapies and frequent recurrence are still major problems. Therefore, understanding the mechanisms of acquired TMZ resistance and identifying new targets in chemoresistance are important for developing effective GBM treatment strategies.

In the present study, we generated a TMZ-resistant glioblastoma cell line with limited proliferative potential and an inhibited apoptotic process when treated with TMZ. Through proteomics and metabolomics analyses, we found that the TCA cycle was inhibited when cells acquired TMZ resistance. The TCA cycle is a central hub that orchestrates energy metabolism and macromolecule synthesis in mitochondria. It produces reducing equivalents in NADH and FADH_2_, biomolecules that are utilized for ATP generation, through oxidative phosphorylation. Mitochondrial respiration analysis showed that oxidative phosphorylation capacity was decreased. These results revealed mitochondrial dysfunction in U87 cells after long-term TMZ treatment.

Mitochondrial mtDNA encode critical proteins of ETC complexes which generate a proton gradient and mitochondrial membrane potential (ΔΨm) vital for mitochondrial function [31]. Significantly, by analyzing the TCGA transcriptome dataset, we found a negative correlation between the malignancy of brain tumors and the expression levels of mtDNA-encoded protein genes (Appendix A). In this present study, we observed reduced mtDNA copy number, disruption of ETC, and depolarized mitochondrial membrane potential in U87 DR cells. This suggested the presence of mitochondrial dysfunction in TMZ-resistant glioblastoma cells. To date, mitochondrial dysfunction has been shown to play an important role in various tumors, including breast, prostate, and colorectal cancers [32,33,34]. Similarly, mtDNA depletion was associated with cancer progression to a more malignant phenotype with adverse prognosis in glioblastoma patients [30,35]. mtDNA copy number has been a novel prognostic biomarker in GBM [36,37]. Additionally, we observed in the present study that rotenone-induced mitochondrial dysfunction promoted TMZ resistance in glioblastoma cells and that the susceptibility to TMZ treatment was increased with the prolongation of rotenone treatment (Appendix A). This result suggested that mitochondrial dysfunction was an underlying mechanism in acquired drug resistance.

Mitochondria are essential in maintaining cellular homeostasis [38,39]. Mitochondrial dysfunction causes deficits in energy supply and macromolecule production, concurrent with the induction of abnormal signaling. Depolarization of mitochondrial membrane potential causes the release of free calcium ions (Ca^2+^) into the cytoplasm from mitochondria [40,41,42]. In the present study, we observed the elevation of Ca^2+^ levels in U87 DR cells. Similarly, increased Ca^2+^ levels were also detected in U87 WT cells with rotenone treatment (Appendix A). And JNK, which could be regulated directly by elevated Ca^2+^ levels [20], was also activated in U87 DR cells. When treated with JNK inhibitor SP600125, U87 DR cells showed a higher susceptibility to TMZ.

JNK regulates important cellular activities, including cell apoptosis and survival [43]. STAT3 is one of the substrates regulated by JNK, and its phosphorylation level mediated by JNK was required for its full transcriptional activation [44,45]. In the present study, we detected the activation of STAT3 in U87 DR cells, and it was inhibited when treated with SP600125. In addition, inhibition of STAT3 activity enhanced the antitumor effect of TMZ against U87 DR cells. STAT3 is a key regulator of cancer cell proliferation and survival. Activated STAT3 upregulates anti-apoptotic proteins, including Bcl-xL and survivin, to protect cells from apoptosis [46]. In the present study, we detected increased mRNA levels of Bcl-xL and reduced apoptosis in U87 DR cells. These results suggest the involvement of JNK–STAT3 pathway activation and apoptosis inhibition in TMZ resistance.

In summary, our study revealed that mitochondrial dysfunction alongside the activation of the JNK–STAT3 pathway were correlated with acquired TMZ resistance in glioblastoma cells. And this study suggests that a combination of TMZ and an inhibitor of JNK kinase or STAT3 is a potentially effective strategy for glioblastoma treatment.

## 5. Conclusions

In conclusion, we generated a TMZ-resistant glioblastoma cell line. Multi-omics analyses demonstrated that glycolysis and the TCA cycle were changed in the TMZ-resistant cells. And energy metabolism analyses suggest that mitochondrial function was disturbed in TMZ-resistant cells. Specifically, both mtDNA copy number and mitochondrial membrane potential were reduced in these cells. Interestingly, we found that mitochondrial dysfunction induced by rotenone also decreased the susceptibility to TMZ in U87 WT cells, which confirms the involvement of mitochondrial dysfunction in acquired TMZ resistance in glioblastoma cells. Furthermore, we showed that JNK–STAT3 pathway activation contributed to TMZ resistance in U87 DR cells. These results suggest combining a JNK inhibitor or STAT3 inhibitor with TMZ as a promising strategy for GBM treatment. A limitation of the study is that our observations were based on the behavior of a single resistant cell line. Therefore, future experiments will be required to verify the conclusions in different models or systems. Furthermore, it would also be interesting to explore whether enhancing the maintenance and repair of mitochondrial function could prevent glioblastoma chemoresistance in future studies.

## Figures and Tables

**Figure 1 biomolecules-13-01408-f001:**
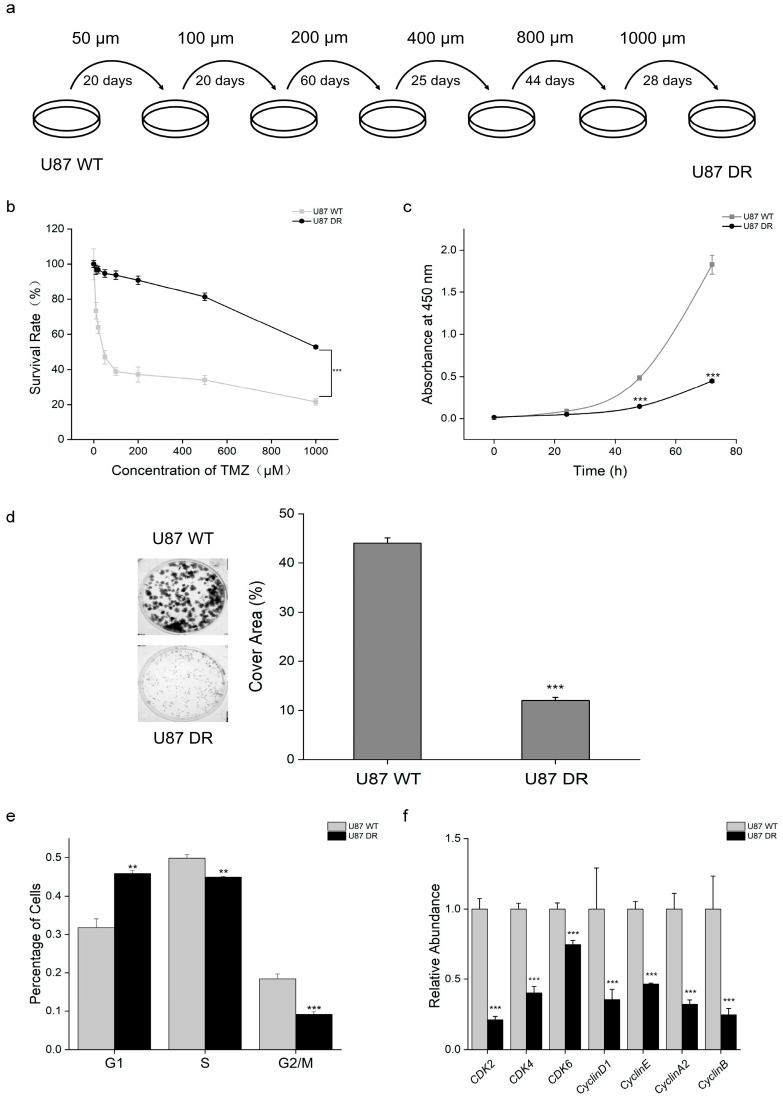
Cell proliferation was inhibited in TMZ-resistant glioblastoma cells. (**a**) Schematic showing the process to established TMZ-resistant glioblastoma cells; (**b**) The viability of U87 WT and U87 DR cells treated with TMZ was assessed by the CCK-8 assay; (**c**,**d**) Proliferative ability was examined using the CCK-8 and colony formation assays in U87 WT and U87 DR cells; (**e**) Analysis of cell cycle by flow cytometry. The acquisition of TMZ resistance induced G1-phase cell cycle arrest; (**f**) mRNA levels of proteins regulating G1/S phase transition were decreased in U87 DR cells as examined through qPCR experiment. Significance was calculated using Student’s *t*-test. ***: *p* < 0.001, **: *p* < 0.01; *n* = 3, mean ± SD.

**Figure 2 biomolecules-13-01408-f002:**
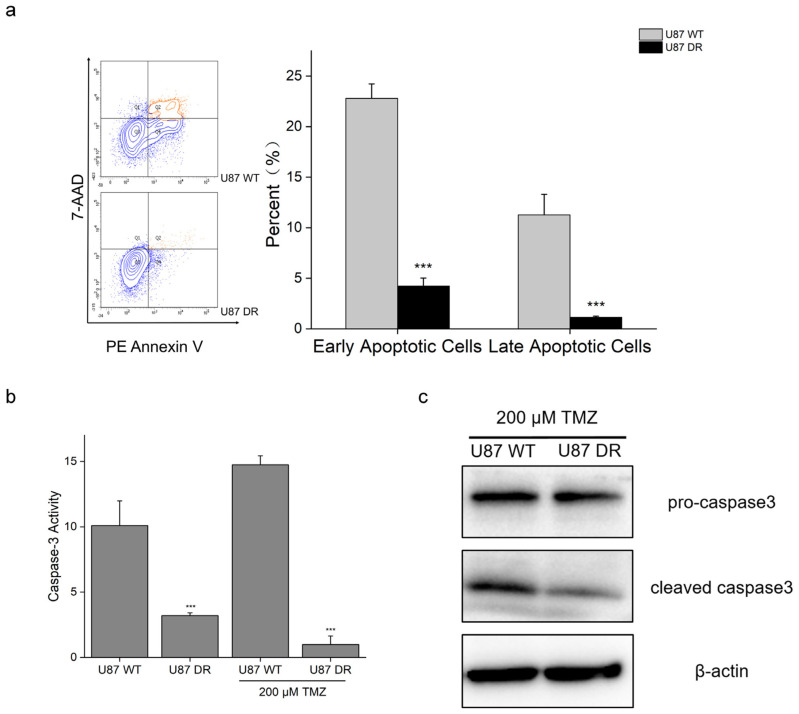
Apoptotic process was inhibited in TMZ-resistant glioblastoma cells. (**a**) PE Annexin V and 7-AAD staining and flow cytometry analysis showed that apoptotic process was reduced in U87 DR cells compared with U87 WT cells when treated with 200 μM TMZ; (**b**) Caspase-3 activity was decreased in U87 DR cells with or without 200 μM TMZ treatment; (**c**) Western blotting results of pro-caspase 3 and cleaved caspase 3 in U87 WT and U87 DR cells treated with 200 μM TMZ; β-actin was used as a control. Significance was calculated using Student’s *t*-test. All original Western Blotting Figures can be find in Appendix A. ***: *p* < 0.001; *n* = 3, mean ± SD.

**Figure 3 biomolecules-13-01408-f003:**
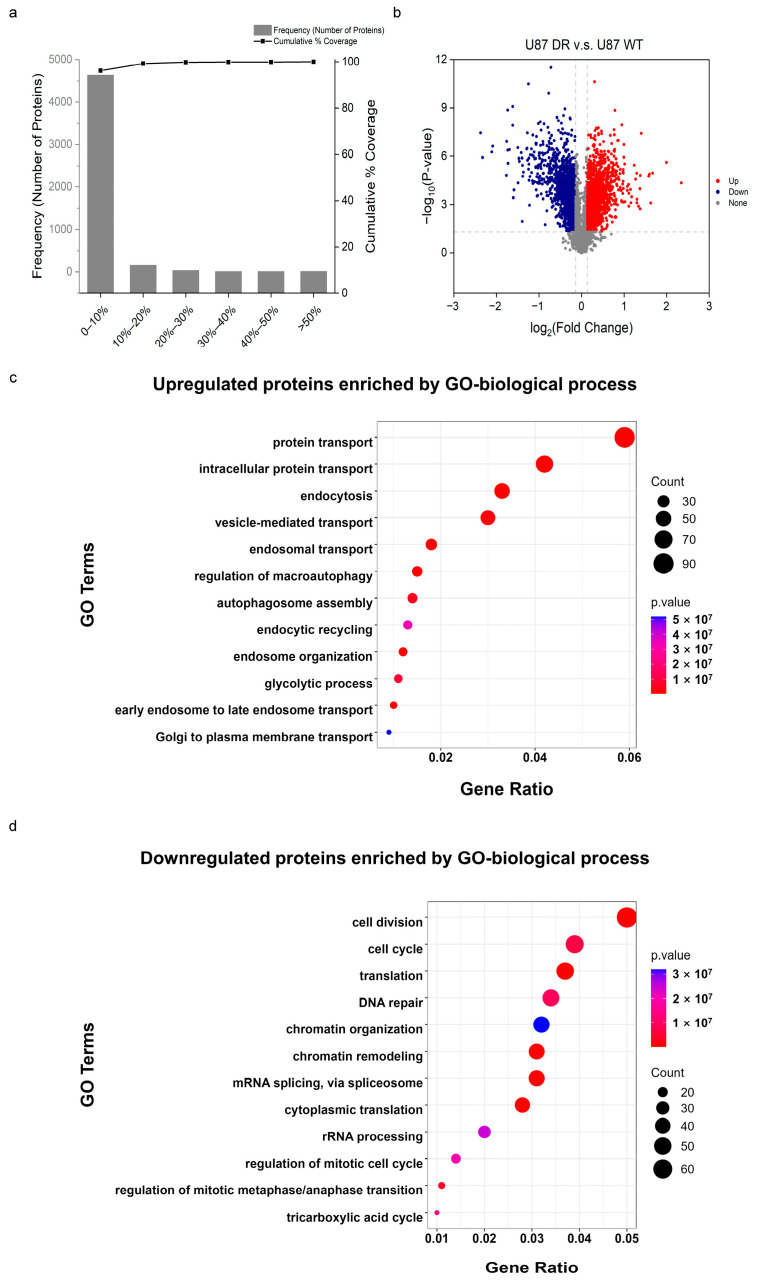
Biological process enrichment of differentially expressed proteins (DEPs) in proteomics analysis of U87 WT and U87 DR cells. (**a**) Experimental variations of proteomics analysis between U87 WT and U87 DR cells; (**b**) Volcano plot of proteins based on Student’s *t*-test *p* values and ratios of protein expressions in U87 WT and U87 DR cells. Blue and red dots indicate the downregulated and upregulated proteins with significant differences (*p* values < 0.05, ratios ≤ 0.9 or ≥1.1), respectively; (**c**,**d**) The biological processes of upregulated and downregulated proteins in U87 DR cells were enriched using DAVID. The size of dots indicates the count of proteins in each term, and color indicates *p* value.

**Figure 4 biomolecules-13-01408-f004:**
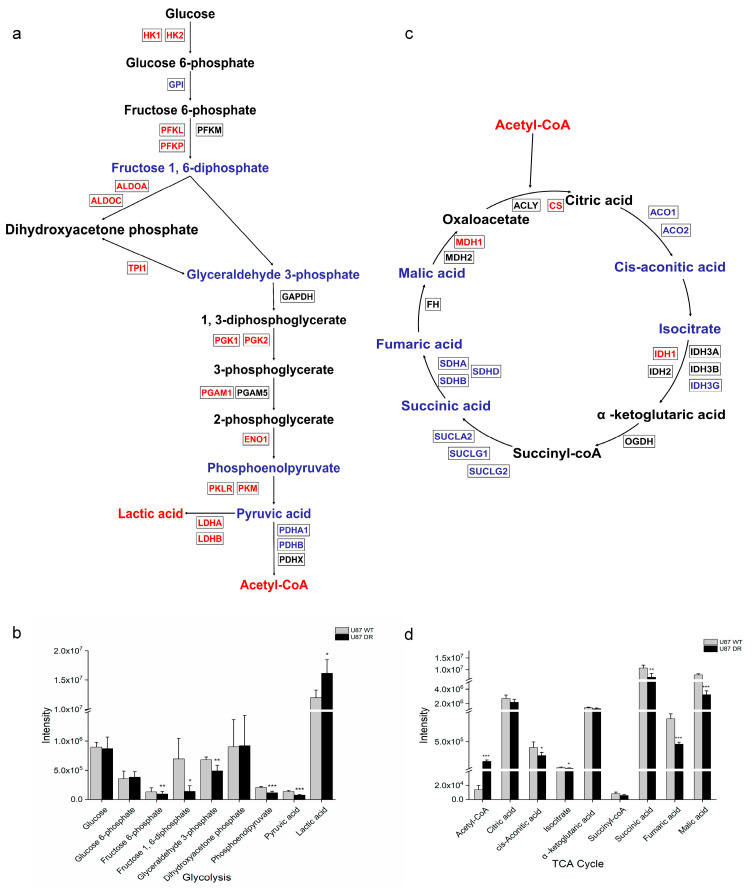
Quantification of metabolites involved in glycolysis and TCA cycle through metabolomics analysis of U87 WT and U87 DR cells. (**a**) Schematic representation of glycolysis in U87 DR cells; (**b**) Quantification of metabolites involved in glycolysis; (**c**) Schematic representation of TCA cycle in U87 DR cells; (**d**) Quantification of metabolites involved in TCA cycle. For (**a**,**c**), upregulated metabolites and proteins are shown in red, while downregulated metabolites and proteins are shown in blue. Significance was calculated using Student’s *t*-test. ***: *p* < 0.001; **: *p* < 0.01; *: *p* < 0.05; *n* = 5, mean ± SD.

**Figure 5 biomolecules-13-01408-f005:**
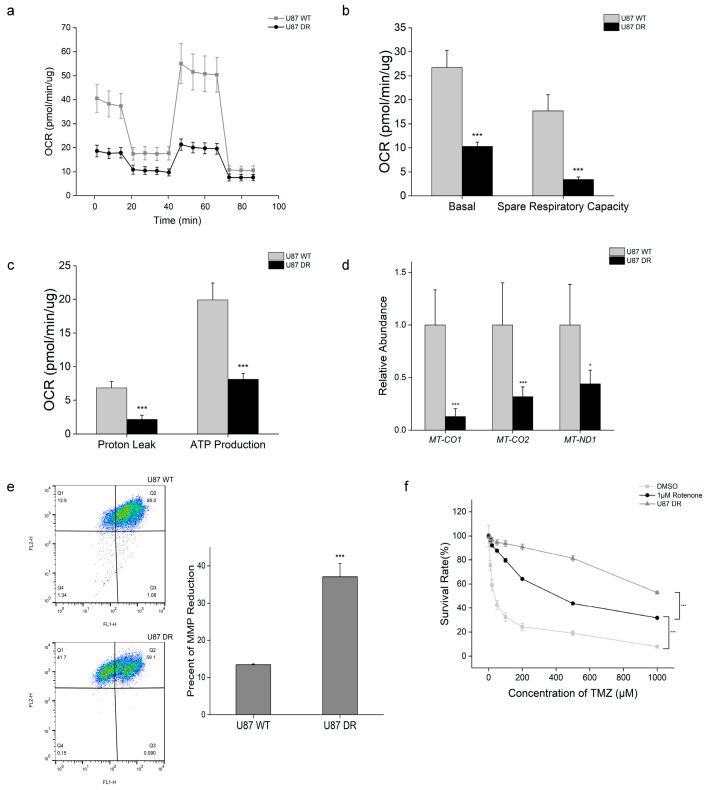
Mitochondrial dysfunction in U87 DR cells. (**a**–**c**) Oxygen consumption rate (OCR) analysis of U87 WT and U87 DR cells; (**d**) mtDNA copy number was determined by qPCR using nuclear and mitochondrial genomic DNA of U87 WT and U87 DR cells. *MT-CO1*, cytochrome oxidase 1; *MT-CO2*, cytochrome oxidase 2; *MT-ND1*, NADH dehydrogenase subunit 1; (**e**) Mitochondrial membrane potential (MMP, ∆Ψm) was examined by JC-10 staining. The fluorescent intensities for JC-10 aggregates and monomeric JC-10 were measured with a flow cytometer using FL1 and FL2 channels, respectively; (**f**) Cell viability of rotenone-treated U87 WT cells was examined by CCK-8 assay. Significance was calculated using Student’s *t*-test. ***: *p* < 0.001, *: *p* < 0.05; *n* = 3, mean ± SD.

**Figure 6 biomolecules-13-01408-f006:**
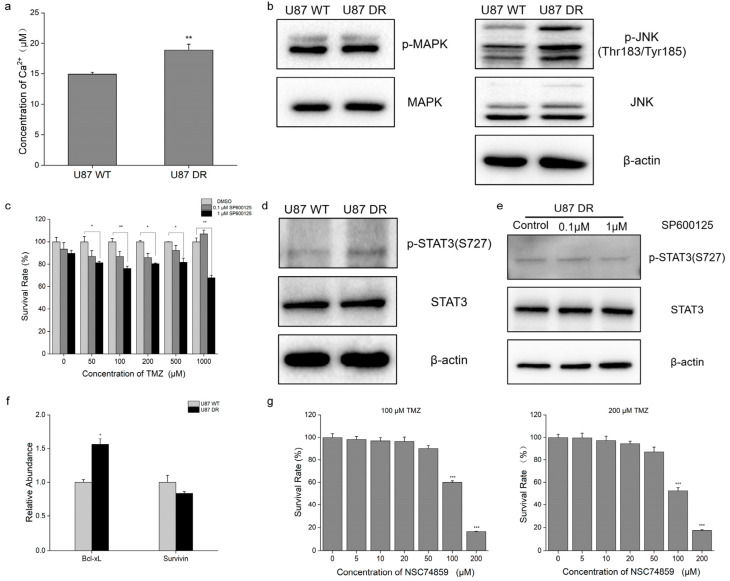
Mitochondrial retrograde signaling was activated when glioblastoma cells acquired TMZ resistance. (**a**) Elevated level of free cytosolic Ca^2+^ was examined in U87 DR cells; (**b**) Western blotting results of mitogen-activated protein kinases (MAPK), phospho-mitogen-activated protein kinases (p-MAPK), c-Jun N-terminal kinase (JNK), and phospho-c-Jun N-terminal kinase (p-JNK); β-actin was used as a control; (**c**) Cell viability upon SP600125 and TMZ treatment was examined using CCK8 assay in U87 DR cells; (**d**) Western blotting results of signal transducer and activator of transcription 3 (STAT3) and phospho-signal transducer and activator of transcription 3 (p-STAT3); β-actin was used as a control; (**e**) Western blotting of STAT3 and p-STAT3 in SP600125-treated U87 DR cells; β-actin was used as a control; (**f**) qPCR examination of mRNA levels of Bcl-xL and survivin in U87 WT and U87 DR cells; (**g**) Cell viability upon NSC74859 and TMZ (100 μM left, 200 μM right) treatment was examined using CCK8 assay in U87 DR cells. Significance was calculated using Student’s *t*-test. ***: *p* < 0.001; **: *p* < 0.01, *: *p* < 0.05; *n* = 3, mean ± SD.

## Data Availability

Mass spectrometry data of proteomic analysis have been uploaded to iProX (iProX ID: IPX0007126000). Mass spectrometry data of metabolomic analysis have been uploaded to MetaboLights (MetaboLights ID: MTBLS8596). The datasets for this study are available from the corresponding author upon reasonable request.

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
