# Peer review of "Multi-Omics Analyses Reveal Mitochondrial Dysfunction Contributing to Temozolomide Resistance in Glioblastoma Cells"

_biomolecules, 2023, doi:10.3390/biom13091408_

Round 1
Reviewer 1 Report
In this particular investigation, Deng and colleagues successfully generated a cell line that exhibited resistance to TMZ-11. Through the utilization of multi-omics analyses and energy metabolism analysis, authors can able to ascertain that mitochondrial dysfunction played a significant role in the development of resistance to TMZ-12 in glioblastoma cells. The administration of rotenone led to alterations in mitochondrial functions, consequently leading to the development of resistance to TMZ in glioblastoma cells. It was discovered through the present investigation that there were heightened levels of Ca2+ and activation of the JNK-STAT3 pathway in cells that exhibited resistance to TMZ. Furthermore, the sensitivity to TMZ was enhanced upon inhibition of JNK kinase and STAT3. Collectively, the findings of this study indicate that the concurrent administration of TMZ 17, along with an inhibitor targeting either JNK kinase or STAT3, holds promise as a viable therapeutic approach for glioblastoma.
The work is generally well written, the techniques are well explained, and the results are thoroughly discussed. Although the study given is not ground breaking, the models offered in the paper are a significant contribution to the knowledge base. The data presented in the paper supports the conclusions. The figures and tables are clear enough and self-explanatory.
Before accepting this article, I have couple of minor concerns to be addressed
1. The authors should add a statement indicating the limitation of the study in the abstract or conclusion section.
2. The authors should present a flow diagram of overall methodology/analysis employed in the manuscript that can help the readers to grasp the overall direction of the paper.
Reviewer 2 Report
1) Literature references should be provided in [1, 2, 3 etc] in brackets 2) U87 WT and U87 DR cells were cultured in Dulbecco’s Modified Eagle Medium (DMEM) medium (Wisent, Nanjing, China). The article must be strictly checked for errors, repeated words. 3) How long the cells were cultured before the experiments. How was the analysis for mycoplasma contamination carried out. From which passage the cells were used for experiments. 4) 2.6. Western Blotting Analysis. Antibody dilution should be submitted 5) 2.13. Concentration of Calcium Detection. Should be rephrased. Example: Determination of the calcium ions concentration in the cytosol 6) Caspase 3 is a prototypical apoptotic executioner that cleaves many other function- 277 ally critical proteins, leading to apoptosis26. Completely incomprehensible sentence. It needs to be rephrased. As well as a number of other sentences. 7) 3.3. Expression of Proteins Involved in Glycolysis and TCA Cycle Changes in U87 DR cells. This section should be expanded 8) Figure 5. The quality of the figure should be improved and the symbols in the figure should be enlarged 9) The conclusion should be written more clearly and strictly summarize the results. The way the conclusion is presented at the moment can be presented at the end of the discussion chapter.
A number of phrases and sentences need to be corrected. The quality of the English language needs significant improvement
Reviewer 3 Report
Zhang et al., present a work where they develop a Temozolomide (TMZ) resistant glioblastoma cell line (U87) and using different approaches claim that part of the resistance mechanism is due to mitochondrial dysfunction. The authors also state that resistant cells show activated JNK-STAT3 pathway and propose that combination of TMZ and an inhibitor of JNK or STAT3. Although interesting, conclusions were based on the behavior of a single resistant cell line and the conclusions about the mechanism of action are over-interpreted.
Major Concerns
1)The paper is based on the development of a cell line that is resistant to TMZ. However, is unclear how the IC50 of both cell lines were calculated. According to the IC50 definition, this value corresponds to the concentration of a specific drug required to produce 50% of maximal inhibition (which is different from 50% of cell viability or 50% of cell mortality). From our understanding, the probit approach transforms the % of cell mortality into probits (probability units), which are graphically represented in function of the drug concentration utilized. Since the linear regression is posteriorly fitted into this data, does this mean that the probit approach does not consider the maximum and minimum effects of this drug in this cellular model? And considers that the maximum and minimum always corresponds to 0% and 100% of inhibition of cell viability, respectively? If this is the case, we are not truly calculating the concentration of this drug that corresponds to 50% of cell viability inhibition.
2) The concentration range utilized does not include the concentration of TMZ necessary to achieve 50% of survival rate in the U87 DR cell line. Assuming that this data points were used to graphically represent the probit inhibition rate, shouldn’t higher TMZ concentrations be included for a better extrapolation of the linear regression, specifically for the U87 DR cell line? We suggest a more extensive method section explaining the calculation of IC50 and how the probit regression was applied to the drug response data.
3) The authors compared colony formation ability and cell cycle analysis in U87 and U87DR counterpart in basal conditions. The analysis of these parameters also in the presence of TMZ would be both of general interest and would be therapeutically informative.
4) The authors should comment on why resistant cells show less colonies and less survival rate.
5) The authors use rotenone as a treatment to induce damage in the mitochondria and mimic the behavior of U87DR cells. Results would be strengthened by including the U87DR cell line and compare effects. It is also unclear the length of the treatment in figure 5f.
6) p.14: Data on Fig. 5 suggest that mRNA levels of core important mitochondrial genes, including PGC1a, are diminished in resistant cells. PGC1a is a major regulator of mitochondrial biogenesis. However, total number of mitochondria remained constant in both cells. It would be important to analyze PGC1a activation status.
7) In Figure 6, the authors try to elucidate the mechanism in U87 DR cells leading to resistance to TMZ. In Figure 6b it is shown that JNK is phosphorylated in U87DR cells. However, analyzing the original blot, total JNK is also elevated, suggesting that the increase in the phosphorylation mark of JNK might be due to an increase in the overall level of JNK and not due to activation.
In figure 6c, the authors show that U87 DR cells present a decreased survival rate in the presence of a JNK inhibitor, concluding that inhibition of JNK restores sensitivity to TMZ. However, this conclusion appears to extend beyond the level of experimental rigor applied to this subject. Increase in cell death might be due to increase toxicity of treating cells with 2 drugs simultaneously and not necessarily due to manipulation of JNK activation status. Similar conclusions are drawn from treatments with STAT3 inhibitors in Figure 6g.
7) In Figure 6e, cells are treated with a STAT3 inhibitor. It is unclear what was the conclusion drawn from this experiment. It would be important also to include the u87 WT cell line.
Minor Concerns
1) It is not clear what is the length of each treatment between each round of drug exposure.
2) It is not stated what is the concentration of TMZ used in figure 1C.
3) p.8: Figure numbering should be reviewed on section 3.2.
4) p.10: The sentence: ”…there was no correlation between autophagy and TMZ-resistance in U87 DR cells” should be carefully worded. It is an overstatement to conclude that there is no correlation between an extremely complex cellular process, as autophagy, based on the data presented.
5) p.14: fig 5e is imperceptible.
Round 2
Reviewer 2 Report
The quality of the article has been improved. The article is ready for publication in its current form